# Visualization of Swift Ion Tracks in Suspended Local Diamondized Few-Layer Graphene

**DOI:** 10.3390/ma16041391

**Published:** 2023-02-07

**Authors:** Nadezhda A. Nebogatikova, Irina V. Antonova, Anton K. Gutakovskii, Dmitriy V. Smovzh, Vladimir A. Volodin, Pavel B. Sorokin

**Affiliations:** 1Rzhanov Institute of Semiconductor Physics of the Siberian Branch of the RAS, Novosibirsk 630090, Russia; 2Department of Semiconductor Devices and Microelectronics, Novosibirsk State Technical University, Novosibirsk 630087, Russia; 3Physical Department, Novosibirsk State University, Novosibirsk 630090, Russia; 4Kutateladze Institute of Thermophysics, Siberian Branch of the Russian Academy of Sciences, Novosibirsk 630090, Russia; 5Technological Institute for Superhard and Novel Carbon Materials, Moscow 108840, Russia

**Keywords:** suspended graphene, high-energy ion irradiation, ion tracks, nanodiamond, internal strain

## Abstract

In the present study we investigated the nanostructuring processes in locally suspended few-layer graphene (FLG) films by irradiation with high energy ions (Xe, 26–167 MeV). For such an energy range, the main channel of energy transfer to FLG is local, short-term excitation of the electronic subsystem. The irradiation doses used in this study are 1 × 10^11^–5 × 10^12^ ion/cm^2^. The structural transformations in the films were identified by Raman spectroscopy and transmission electron microscopy. Two types of nanostructures formed in the FLG films as a result of irradiation were revealed. At low irradiation doses the nanostructures were formed preferably at a certain distance from the ion track and had the form of 15–35 nm “bunches”. We assumed that the internal mechanical stress that arises due to the excited atoms ejection from the central track part creates conditions for the nanodiamond formation near the track periphery. Depending on the energy of the irradiating ions, the local restructuring of films at the periphery of the ion tracks can lead either to the formation of nanodiamonds (ND) or to the formation of AA’ (or ABC) stacking. The compressive strain value and pressure at the periphery of the ion track were estimated as ~0.15–0.22% and ~0.8–1.2 GPa, respectively. The main novel results are the first visualization of ion tracks in graphene in the form of diamond or diamond-like rings, the determination of the main condition for the diamond formation (the absence of a substrate in combination with high ion energy), and estimates of the local strain at the track periphery. Generally, we have developed a novel material and have found how to control the film properties by introducing regions similar to quantum dots with the diamond interface in FLG films.

## 1. Introduction

Diamond, due to its superior properties, is quite attractive to both the fields of science and technology. However, the thermodynamic instability of the diamond raises questions about the sustainability of its structure especially at the nanolevel and the phase transformation between graphite/graphene and diamond/nanodiamond (ND), e.g., diamonds are thermodynamically unstable when heated to 1800–2200 K and are subject to the graphitization process [1].

The formation of natural diamond requires high pressures (70–130 GPa) and temperatures (~3300–7300 K) [2]. Another important condition for the formation of natural diamond is a rapid jump or release of pressure, similar to what happens in a kimberlite volcano (pipe) [3]. Moreover, even if the diamond phase has formed, to maintain it in a stable state, it is necessary to comply with a certain set of conditions for the pressure and temperature. Another type of diamond, also found in nature, is the impact diamond, formed at the sites of meteorite impacts or in the meteorites themselves. Their defining features are the presence in their structure of a specific hexagonal diamond phase (lonsdaleite, wurtzite structure) [4,5] and a size up to 1–100 µm. In [6] an estimation was made for the nanodiamond mass fraction for primitive carbonaceous meteorites, which equaled ~300 ppm.

Conditions, similar to the fall of a meteorite, can be recreated by explosions, high-energy laser impulses [7], cosmic rays or swift ion irradiation [8]. The typical range of diamond sizes is in the nanometer range, so they are called nanodiamonds. During the explosions and laser irradiation the temperature can rise up to 3000–4000 K and the pressure up to 15–30 GPa. NDs created by such a process are called detonation nanodiamonds (DND), and they are typically several nanometers in size. Inside the DND there is often a sp^3^-hybridized core 4–6 nm in size and the following shell usually has a thickness up to 1 nm and consists of sp^2^-hybridized carbon atoms formed by the surface graphitization effect. The outer layer can contain various functional groups and heteroatoms. The DNDs often involve inclusions and defects and their diameters are rarely larger than 10 nm. However, these DNDs are very attractive for bio-imaging and drug delivery due to their small size [9]. Despite almost half a century of DND research, there is still no single point of view among researchers on the process of their formation [10].

To implement the conditions and study the mechanism of cosmic ray exposure, high-energy ions obtained at accelerators can be used [11]. Previously, studies have already been carried out on the irradiation of graphite and graphene films with charged particles with different energies (electrons, H+ ions, swift heavy ions, etc.) [12,13,14,15]. In [16] it was shown that when a graphite foil 125 μm thick is irradiated with Kr ions with an energy of 400 MeV and a dose of 6 × 10^12^ ion/cm^2^, NDs with a characteristic size of 4 nm are formed in the foil. The efficiency of their formation is ~0.08 diamond/ion.

It seems attractive to realize a controlled phase transition when local influence on the graphite structure induces its sp^3^-hybridization. Thus, previously it was shown that direct electron irradiation with a low energy of the hydrogenated carbon films allowed the formation of both isolated nanodiamond clusters [17] and 2D diamond areas [18]. It seems interesting to try to realize such an effect by the exposure to high-energy ions that allow for complete rearranging of the structure of the material. Here, we studied such a process, i.e., the local rearrangement of the structure of graphene layers under the action of swift heavy ions. The process of nanosized diamond (ND) formation using high-energy laser irradiation was theoretically studied in [7]. The probability of ND formation under the most favorable conditions was estimated as 10^−4^–10^−3^.

Diamane-like structures represent a new class of 2D materials [19,20]. The unique properties of 2D diamonds are the high thermal conductivity, high hardness, optic peculiarities, and wide-gap spectra with many resonance peaks. They make such structures promising candidates for the components in future nanoelectronics. Graphene/nanodiamond-based heterostructures, superlattices, nanomeshes, and composites are a basis for construction of the novel materials and applications [21,22,23,24]. Such structures make it possible to control the structural properties, the energy gap, and often to give the material new properties that cannot be obtained by any other means.

Previously, we studied two types of films nanostructured by irradiation. They were electrostatically exfoliated few-layer graphene and CVD few-layer graphene and were transferred onto oxidized silicon substrates [25,26]. We have found that in locally suspended regions (folds, bubbles) the process of rearrangement of the film structure proceeds most noticeably. Therefore, in the present study, we created and studied suspended CVD graphene films on TEM grids. It has been shown *in situ* that when using Xe ions with an energy of ~26 and 167 MeV and at relatively low irradiation doses, the rings of nanodiamonds with typical diameters of 15–35 nm are formed around ion tracks in irradiated films. The main highlighted results are the first visualization of ion tracks in graphene in the form of a diamond, the determination of the main condition for diamond formation (absence of a substrate), and estimations of the local strain value at the track periphery (~0.2%).

*In situ*, it has been shown that when Xe ions with energies of ~26 and 167 MeV are used with relatively low radiation doses, rings of nanodiamonds with a characteristic diameter of 15–35 nm are formed around ion tracks in irradiated films. The main results are the first visualization of ion tracks in graphene in the form of a diamond, the determination of the basic condition for diamond formation (absence of a substrate), and estimation of the magnitude of the local deformation at the periphery of the track (~0.2%).

## 2. Materials and Methods

The graphene synthesis on a copper foil was carried out at the temperature of 1070 °C in the Ar–H_2_–CH_4_ working gas mixture flow [27]. The thickness of the grown films was estimated up to ~9 monolayers (up to 3 nm). Thus, we had few-layer graphene (FLG) areas in the grown films. Then, FLG was transferred onto the desired substrate (SiO_2_/Si or specialized grid for transmission electron microscopy) using the Cu etching 0.1 M solution of (NH_4_)S_2_O_8_ and a wet transfer.

The samples were subjected to the Xe ion beam with the ion energies 26 and 167 MeV at room temperature in a vacuum at the pressure of 6.3 × 10^−6^ Torr. To avoid the graphene film heating, the ion flux density was limited to the range of (2.0–5.7) × 10^8^ cm^−2^s^−1^. The irradiation doses ranged from 1 × 10^11^ to 5 × 10^11^ ion/cm^2^. We tried to use the ion doses below the value of the ion track overlapping for the individual track observation. For many materials it is ~10^12^ ion/cm^−2^. Graphene has a high thermal conductivity, so we used the dose of 5 × 10^11^ ion/cm^−2^ and lower. The dose of 5 × 10^11^ ion/cm^−2^ corresponds to the average distance between the ion tracks of 14 nm. The irradiation was performed on the ion beamline at the IC-100 cyclotron of FLNR JINR (Dubna). We used ~4–5 samples for each ion energy and dose. The total quantity of samples was 28 samples.

The structural properties of pristine (non-irradiated) and irradiated films were examined using the following experimental techniques: transmission electron microscopy (TEM) and Raman light scattering. The Raman spectra were recorded in the back-scattering geometry with the use of a triple T64000 Horiba Jobin Yvon spectrometer. Raman spectra were obtained at room temperature using the GFL-515-0200-FS ytterbium fiber laser (the 514.5 nm excitation line) and at low power (1.3 mW). To avoid local heating during the measurements, the diameter of the laser beam on the samples was increased to 10 μm. To determine the parameters and ratio of the peaks, the deconvolution of the spectra was carried out using the Fityk program [28]. High-resolution transmission electron microscopy (HRTEM) measurements were carried out on a TITAN 80-300 cubed (FEI) electron microscope equipped with a Cs lens spherical aberration corrector at an accelerating voltage of 300 kV. The digital processing of experimental HRTEM images was carried out using the commercial GMS-2.32 (GATAN) software package and the free Gwyddion software [29].

## 3. Results

In the present study, FLG films were transferred onto special TEM grids serving as substrates. The sketch of the sample preparation is shown in Figure 1. As a result, we had local areas of suspended FLG, which were then irradiated, and they were observed using TEM. Figure 2 presents the HRTEM images of nanosized objects embedded in the FLG films irradiated by Xe ions with energies 167 and 26 MeV. Information about the interlayer distance values for the lines 1–7 is shown in the table in Figure 2e.

The formation of nanostructured objects with characteristic lateral sizes ranging from a few to tens of nm was found in the FLG films after ion irradiation. The TEM images showed that often such regions were formed not separately, but clustered as in Figure 3. Apparently, they were located in an area that can be interpreted as the periphery of the ion track. The typical diameter of such an area was ~15–20 nm for 26 MeV and 20–35 nm for 167 MeV Xe ions. Clusters were more distinctive and better visible for the lower irradiation dose (1 × 10^11^ ion/cm^2^). It seems that the high irradiation dose (5 × 10^11^ ion/cm^2^) was characterized by the overlapping of ion tracks resulting in the stressed regions of the clusters relaxing under due to the energy received from the next irradiating ions, rearranging the atoms into less stressed configurations such as that in Ref. [30].

At the irradiation dose of 1 × 10^11^ ion/cm^2^ and 167 MeV Xe ions, TEM images mainly showed nanoobjects with a more complex structure compared to that found for 26 MeV ions. After irradiation of the films with 167 MeV Xe ions, nanocrystals with clearly defined faces, a sixth-order symmetry axis, and the twinning effect were formed more often in comparison with the 26 MeV Xe ions irradiation result. Nanostructured areas with one distinguished direction were more common for the 26 MeV-irradiated films. We revealed the more complex nanoobjects as nanodiamonds and the simpler ones as a local shift of the lattice layers (similar to the local formation of metastable AA, AA’, or ABC regions). It should be noted that even for an ion energy of 167 MeV, AA’-like regions are sometimes encountered. However, a strong dependence of their quantity on the radiation dose was observed. Most of them were found at a lower irradiation dose (1 × 10^11^ ion/cm^2^).

Figure 4 shows histograms based on the analysis of TEM images. The large number of interlayer distances observed in the TEM images can indicate the formation of lonsdaleite (wurtzite/hexagonal diamond) or twinned nanodiamonds in the films similar to results in [4,16,31]. Figure 4a–c shows the characteristic sizes of the nanostructured regions. It should be noted that Figure 4c is characterized by the presence of regions with large diameters (more than 20–25 nm). Figure 4d–i show histograms for the interlayer and interplanar spacings, which were either estimated directly from TEM images (d–f) or by fast Fourier transformation (FFT) analysis. The area under the histograms corresponds to ~100–130 d_i_ values for every set of irradiation conditions. The reason for the difference between the data for the direct measurements and for the measurements based on the length of vectors in reciprocal space is the presence of data not only on interlayer distances but also on in-plane distances between the atomic lines.

A comparison of the obtained histograms shows that for a dose of ~5 × 10^11^ ion/cm^2^, the characteristic sizes of nanostructured regions were smaller than for a dose of 1 × 10^11^ ion/cm^2^. It is possible that as the irradiation dose increased the critical value of the size of the newly formed phase decreased, i.e., the effective potential barrier between the initial phase and others (AA’, nanodiamond, etc.) decreased, i.e., there is a reduction in the effective potential barrier from the initial phase.

Based on the interlayer distances at an irradiation dose of 1 × 10^11^ ion/cm^2^ and an ion energy of 167 MeV, it can be argued that NDs formed in the FLG films with a density of ~(15–30)% of the ion fluence. The ion energy of 26 MeV is close to the threshold for ND formation, and ND density is ~5% of the ion fluence. Possibly, upon irradiation with the 26 MeV ions, the transition of the graphene layers to NDs occurs in the most mechanically stressed regions only. The other crystallites revealed after irradiation were structured with the intermediate or metastable state, for example, the AA’-packing of layers (where each graphene plane is shifted by a 1/2 hexagon from zigzag AA stacking or by a 1/4 hexagon from armchair AB stacking). The most typical interlayer distances for AA’ stacked graphene layers are 0.213–0.215 nm for d_100_, 0.246 nm for d_010_, 0.18 nm for d_102_, 0.125 nm for d_110_, and 0.11 nm for d_020_ [32]. Another typically observed interlayer distance of ~0.29 nm can be attributed to the NDs [33]. We found such distances in the studied films as it can be seen in Figure 4d–i.

Figure 5 demonstrates the HRTEM images, their FFT patterns, and the interlayer distances for the FLG irradiated with ion energies of 167 and 26 MeV. The colored figures label the reflexes in the FFT patterns and regions in the TEM images related to them. Different pairs of the reflexes in colored circles in Figure 5b are described by the same pairs of the length, i.e., they correspond to planes with a close value of the interlayer distance and orientation. The small angle of the misorientation indicates a slight difference in the normal directions to these planes. A similar situation occurs in Figure 5e where groups of close reflexes are shown inside green and pink circles. The regions in red and pink circles in Figure 5a can be attributed to nanotwinned NDs [34].

There are some planes with a distance of ~0.37 nm between them in the bottom part of Figure 5d. They are supposed to be vertically oriented and expanded graphene layers. They could appear due to local film bending or scrolling, so they are locally mechanically stressed. The irradiation caused interlayer bonding and local lattice restructuring, so there is a mix of graphene/graphite and diamond interlayer distances in the tables in the Figure 5c,f (0.13–0.14, 0.18–0.19, 0.21–0.22, and 0.37 nm). The reflexes for the interlayer distance of ~0.13–0.14 nm, labeled by the cyan squares in Figure 5d, look very weak because they are related to tiny fragments of TEM images. The planes in the pink circle in Figure 5d are tilted, closely resembling the model of graphite to diamond transformation as described in [35] and involving ABC-stacking as one of the stages. Planes, located in the pink square in Figure 5d, can be referred to as a local core of hexagonal diamondization due to the interlayer distance of ~0.19 nm.

In Figure 6, there is a TEM image for a multilayer vertically oriented tube or shell around a crystallized core (interlayer distances ~0.19–0.21 nm, there is a 3rd order symmetry axis). The interlayer distance for the shell of the observed structure (along the white lines) is 0.34–0.35 nm. The bending radius of the outer layers is from 1 to 5 nm. This fact indicated the presence of high mechanical stresses in the observed structure, especially in the inner layers. The interlayer distance value as well as their high curvature suggest that the shell consisted of sp^2^-hybridized carbon. We can speculate that such structures may be formed by the graphitization of nanodiamonds formed in the FLG under the action of an electron beam during TEM measurements. This is consistent with the data from [34,36,37] about the graphitization of NDs and their transformation into nano-onions that proceeds from the outer layers to the inner ones. The authors of [37] suggest a mechanism for the splitting of one diamond layer into two graphene layers due to the difference in density between diamond and graphite. The authors also point out that the presence of defects in the diamond can significantly accelerate the process of structure restructuring. The opposite process took place in [38] for the inversion of stability conditions between NDs and sp^2^-bands, possible under the electron beam irradiation with the growing of ND-core inside the graphite particles. A similar combination of the local ND-core and graphite part is shown in Figure 5d.

Raman spectra in Figure 7a show that the suspended films were similar to the films transferred onto the SiO_2_/Si substrates. For the ion energy of 26 MeV at an irradiation dose of 2 × 10^11^ ion/cm^2^, the spectra of CVD graphene films on the SiO_2_/Si and without a substrate were very similar. For the same irradiation dose and 167 MeV Xe ions, the appearance of asymmetry and a shoulder at the G-peak was observed, associated with the appearance of an amorphous phase and the formation of locally stressed regions in the films. Figure 7b shows the spectrum decomposition for the FLG film irradiated with 167 MeV Xe ions at a dose of 5 × 10^12^ ion/cm^2^. Usually, the ratio of D and G (I_D_/I_G_) peak intensities is used for Raman data interpretation with the peak decomposition. However, if the spectra contain several peaks at once, as in our work, the fraction of each peak is often estimated based on the area under them. To determine the parameters and ratio of the peaks, we deconvoluted the spectra using the Fityk software [28]. The results of the peak decomposition are shown in Table 1. Surprisingly, the D-peak area slightly decreased after the irradiation. It is supposed that this was due to the appearance of amorphous carbon modes of both the sp^2^ and sp^3^ hybridization, defects, and heteroatoms (total area sum is ~43%).

## 4. Discussion

The theoretical study of the ND formation process under the high-energy laser irradiation demonstrates that the maximum probability of ND formation at the most favorable conditions can be estimated as 10^−4^–10^−3^ [7]. In our previous study of irradiation with fast heavy ions, the ND formation probability was 2–3 orders of magnitude higher depending on the substrate influence. The ratio of the nanodiamond density to the ion irradiation fluence was ~10^−1^ diamond/ion for areas on the substrate and ~0.8–1 for locally suspended areas [25,26]. A similar value was observed in [16] for graphite, which equaled 0.08 diamond/ion. We transferred the films onto TEM grids in the current study in order to implement conditions for the efficient formation of pores in locally suspended FLG. However, the ratio of the nanodiamond density to the ion irradiation fluence, obtained in the current study, was ~10^−1^ diamond/ion again.

The Raman spectrum of natural diamonds and detonation NDs, with diameters larger than 10 nm, usually contains sharp peak at ~1332 cm^−1^ [9]. The Ref. [40] argues that nanopowders with an ND content less than 30% do not have a diamond-related peak at 1326–1332 cm^−1^. Furthermore, the Ref. [41] shows that for ND sizes about 3–5 nm the meteorites also have no peaks in the 1326–1332 cm^−1^ region, but D and G bands are observed. The studied films with embedded nanodiamonds fit both of the above conditions. However, it is possible that it made sense to use a UV laser with a shorter wavelength for excitation when recording Raman spectra.

At small ND sizes of 3–5 nm, the Raman spectra were close to that demonstrated in the current study and for the example in [14]. In that paper, an analysis of the surface of graphite irradiated with Ar^8+^ ions with an energy of 800 eV (the irradiation dose in [14] is unknown) reported the formation of nanodiamond clusters and a large amount of amorphous carbon. Carbon amorphization is supposed to be an important part of the model of diamond formation, which is a stabilizing layer between the graphite matrix and ND crystallites. According to [14], a droplet of amorphous carbon can be transformed by additional heating into nanodiamonds with sizes up to three nm because they are energetically more favorable than crystallites with sp^2^ hybridization [42].

Surprisingly, the D-peak area in the Raman spectra was slightly decreased after irradiation. This can be partly explained by the appearance of additional peaks associated with the amorphous carbon phase. We found a G-peak position shift between the virgin and irradiated films. The largest value was obtained for the irradiation dose of 2 × 10^11^ ion/cm^2^ with the 26 MeV Xe ions, which equaled −(7–11) cm^−1^. We used data from [43,44] about the relationship between the strain-induced phonon shift and pressure for a five-layered graphene film to estimate the mechanical strain and the pressure that causes it, which were ~0.15–0.22% and ~0.8–1.2 GPa, respectively.

Most of the NDs in TEM images for the dose of 1 × 10^11^ ion/cm^2^ looked assembled in clusters as if they were formed at the ion track periphery. The typical ion track diameter extracted from the ND distribution was about 15–35 nm. The average ratio between the dose and the ND cluster density was about 10%. This was much lower than the pore density observed for the locally suspended regions in irradiated CVD graphene on SiO_2_/Si substrates [25]. It seems that the effective nanostructuring can be explained either by the influence of the substrate (reference atoms ejected from the substrate and embedded in the carbon lattice can hinder the phase transition) or by the impact of the mechanical strain in the films. Indeed, the formation of a denser diamond in a multilayered graphene causes stress in its lattice. The suspended film is able to reduce the out-of-plane stress more effectively, in contrast to the film on the substrate, and the existing mechanical stress in the graphene film reduces the influence of the in-plane stress, whose energy is proportional to the graphene elastic constant [45]. Such an assumption is supported by the fact that for the irradiation dose of 5 × 10^11^ ion/cm^2^ we observed in TEM images a large number of nanodiamonds near the film edges. It is also possible that for the formation of nanodiamonds it is important to damage or deform/strain the graphene layers to create nucleation centers for diamondization [46].

Comparing the TEM for the irradiation doses of 1 × 10^11^ and 5 × 10^11^ ion/cm^2^ (167 MeV Xe ions), it is remarkable that at a dose of 5 × 10^11^ ion/cm^2^ almost all the newly formed structures can be referred to nanodiamonds by the interplanar distances and geometry. Apparently, this was due to the fact that as the irradiation dose increased, some changes accumulated in the structure of the multigraphene films, which facilitated the transition to the nanodiamond phase. At a dose of 1 × 10^11^ ion/cm^2^, the arrangement of nanodiamonds more resembled the edges of a miniature volcano with an amorphous central part instead of a vent, while at the dose of 5 × 10^11^ ion/cm^2^ their arrangement was more homogeneous (Figure 3 and Appendix A).

The formation of facets in some NDs in films irradiated with 167 MeV ions indirectly indicates that the films were locally heated higher when irradiated with 167 MeV ions than when irradiated with 26 MeV ions. At scales of a few nanometers, the formation of facets and different ND shapes can indicate thermodynamically different structuring conditions that occurred depending on the irradiating ion energy [47].

We assumed that the restructuring of the studied graphene layers goes through the local melting and formation of amorphous carbon droplets. For smaller irradiation doses (1 × 10^11^ ion/cm^2^), the additional pressure generated by the internal mechanical stress or/and the passage of the compression wave through the material from the center of the ion track was often required for lattice restructuring. Meanwhile, for high irradiation doses (5 × 10^11^ ion/cm^2^), when a number of structural defects had already accumulated in the material, the probability of a transition from graphene to ND increased and could occur without additional pressure. Apparently, when the films were irradiated, radial compression waves emanated from the ion trajectory, which at a certain distance from the center of the ion trajectory, together with an elevated temperature created conditions for the phase transition of graphene layers into various forms of carbon. A similar pattern of cluster formation around one center is more typical for insulators than for metals [48]. It is known that nuclear collision cascades are capable of inducing a sound/shockwave in the materials. Such a shock wave can in principle cause the emission of material from the impact site, especially for thin films. At the same time, a pressure wave or long-term stress relaxation causes plastic deformation of the material outside, also far from the region of the atomic collisions [48].

During the transition of graphite to diamond, the high temperature in combination with mechanical stresses will accumulate in the lattice of the material, which will significantly affect the further process of structural rearrangement. The strong local heating of graphene layers (up to 3000–7000 °C) takes place in the picosecond time-scale (10^−12^–10^−11^) [26,49]. The characteristic size of the regions formed around the transition centers and yet able to remain stable after the transformation depends strongly on the applied pressure and is inversely proportional to it [45,50]. In addition to mechanical stresses, other factors may also play a role in nanostructuring processes e.g., the size of formed nanodiamonds can be controlled by stabilizing the dangling bonds [42].

The analysis of the obtained experimental data (TEM microscopy, Raman spectroscopy) showed a strong dependence of the nanostructures observed after irradiation on the ion dose and energy. Based on the characteristic size and shape of the Raman spectra, we can conclude that the NDs obtained in the current work are closer to the ultrananocrystalline (3–5 nm) detonation NDs, meteorite NDs, or to the NDs obtained during the laser irradiation than to those that are grown by the CVD method. In addition, apparently, an important factor in the formation of the investigated NDs was the use of locally suspended films of CVD graphene.

We obtained the structures that cannot be created except by irradiation with high-energy ions (low-layer graphene with nanodiamond rings). The details of the structures of graphene/nanodiamond boundaries and layers properties have not yet been sufficiently studied and understood, but such structures can be considered as a way of bond reconstruction between the layers at these interfaces. These carbon structures can be considered as superlattices of joined graphene quantum dots (or, in some cases, nanotubes) and few-layer graphene fragments separated by nanodiamonds or AA’ (or ABC) stacking areas. Their stability and properties were the purpose of the study. As it was demonstrated, for example, in the theoretical studies of Chernozatonskii [19,20], similar materials have great prospectives for applications. The electronic band structure of the theoretically predicted nanomesh can have characteristics that vary from metallic to semiconductor characteristics, depending on the interphase structure, size, shape, and distance between the 2D diamond-like inclusions. The detailing of the electrical properties of the created nanomesh materials will demonstrate the prospectives of the applications.

## 5. Conclusions

We investigated the local changes in the structure of suspended few-layer graphene films as a result of their irradiation with high-energy ions. The structure of the films was analyzed by the means of Raman spectroscopy and transmission electron microscopy. The absence of a substrate had a significant effect on the processes of local nanostructuring of the films. An internal mechanical strain was found in the irradiated films. The value of the compressive strain was estimated on the basis of the G-peak position shift and it was equal ~0.15–0.22% (local pressure ~0.8–1.2 GPa) for the G-peak shift of –(7–11) cm^−1^ in the case of the irradiation with a dose of 2 × 10^11^ ion/cm^2^ and with the 26 MeV Xe ions. The observed rearrangements in the film structure were found to depend on the energy of the irradiating ions. It was shown that some of the formed nanostructures could be interpreted as nanodiamonds. They were often observed in the cases of high ion energies (Xe, 167 MeV). For the lower energies (Xe, 26 MeV) the local restructuring was interpreted as a local shift of the graphene layers relative to the formation of AA’- or ABC-type metastable regions. The large number of interlayer distances observed in the TEM could indicate the formation of lonsdaleite (wurtzite/hexagonal diamond) in the films. The nanodiamonds or AA’- and ABC-type metastable regions formed in suspended FLG films circled with size of 15–20 nm for 26 MeV and 20–35 nm for 167 MeV Xe ions. We assumed the obtained size value is the lateral size of ion tracks and that all nanoobjects are formed at the periphery of the ion track area. It was shown that with the increase in the irradiation dose, the character of spatial distribution of the formed ND becomes more homogeneous due to interactions between tracks. It is shown that for the nanodiamonds formed in the FLG layers, their transition into nano-onions or vertically oriented CNTs under the action of the microscope electron beam is possible.

## Figures and Tables

**Figure 1 materials-16-01391-f001:**
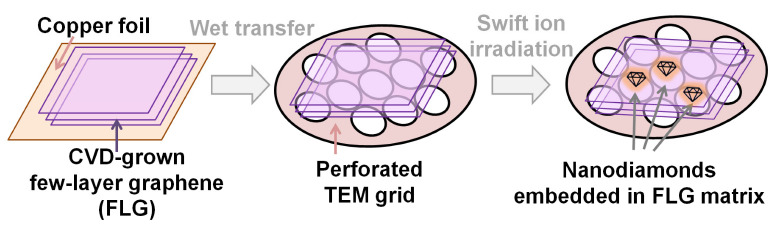
The stages of the sample preparation. Few-layer graphene (FLG) films grown on Cu substrate were transferred onto specialized transmission electron microscopy (TEM) grids. The samples were nanostructured by irradiation with high energy Xe ions.

**Figure 2 materials-16-01391-f002:**
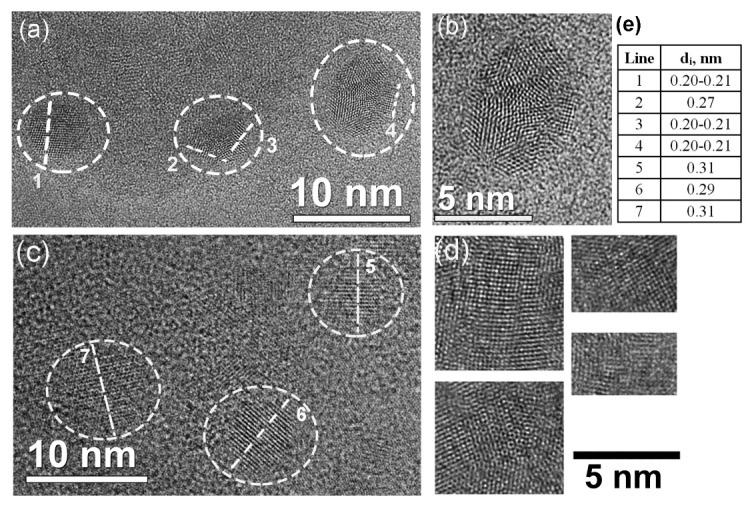
High-resolution transmission electron microscopy (HRTEM) images of nanosized objects embedded in FLG films irradiated with Xe ions. The ion energy was 167 and 26 MeV for (**a**) and (**c**), respectively. The irradiation dose was the same for (**a**) and (**c**), which was 5 × 10^11^ ion/cm^2^. The enlarged image of the top left part of (**a**) is shown in (**b**). Enlarged fragments of the TEM images for the FLG films, irradiated with 26 MeV Xe ions at the dose 1 × 10^11^ ion/cm^2^, are shown in (**d**). Information about the interlayer distance values for lines 1–7 is shown in the table in (**e**).

**Figure 3 materials-16-01391-f003:**
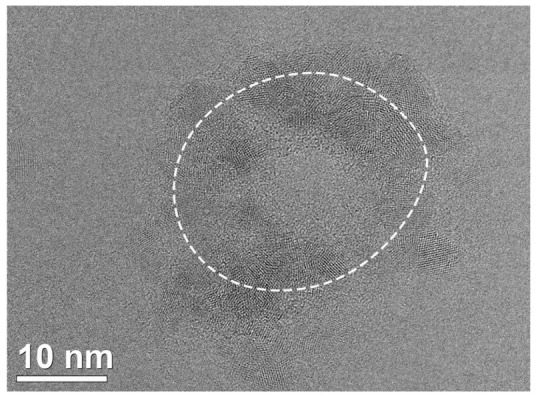
The cluster of nanocrystals at the periphery of the ion track at the irradiation dose of 1 × 10^11^ ion/cm^2^. The ion energy is 167 MeV.

**Figure 4 materials-16-01391-f004:**
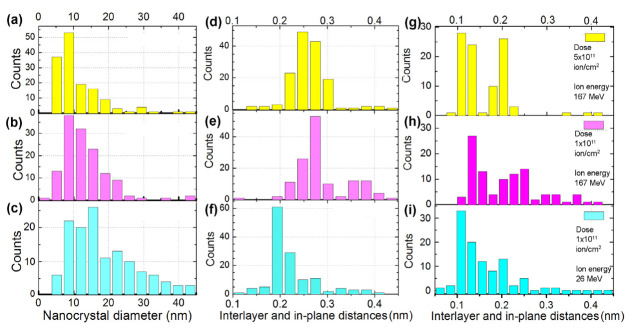
The average nanocrystal diameter (**a**–**c**), and interlayer and in-plane distance (**d**–**i**) distributions for the FLG CVD-films irradiated with Xe ions. The ion energy is 167 MeV for (**a**,**b**,**d**,**e**,**g**,**h**) and 26 MeV for (**c**,**f**,**i**). For (**a**,**d**,**g**) the dose is ~5 × 10^11^ and ~1 × 10^11^ for (**b**,**c**,**e**,**f**,**h**,**i**).

**Figure 5 materials-16-01391-f005:**
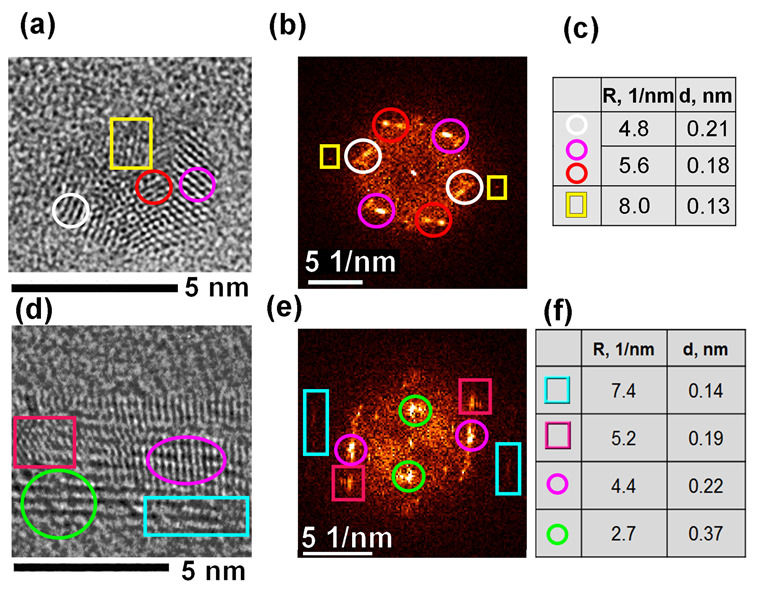
Enlarged fragments of the HRTEM images for CVD FLG films irradiated with 167 (**a**) and 26 (**d**) MeV Xe ions; (**b**) and (**e**) are their fast Fourier transformation (FFT) patterns, respectively. Some bright spots, labeled by colored circles and squares, are described in the tables (**c**) and (**f**). The irradiation doses were 5 × 10^11^ and 1 × 10^11^ ion/cm^2^ for (**a**) and (**d**), respectively.

**Figure 6 materials-16-01391-f006:**
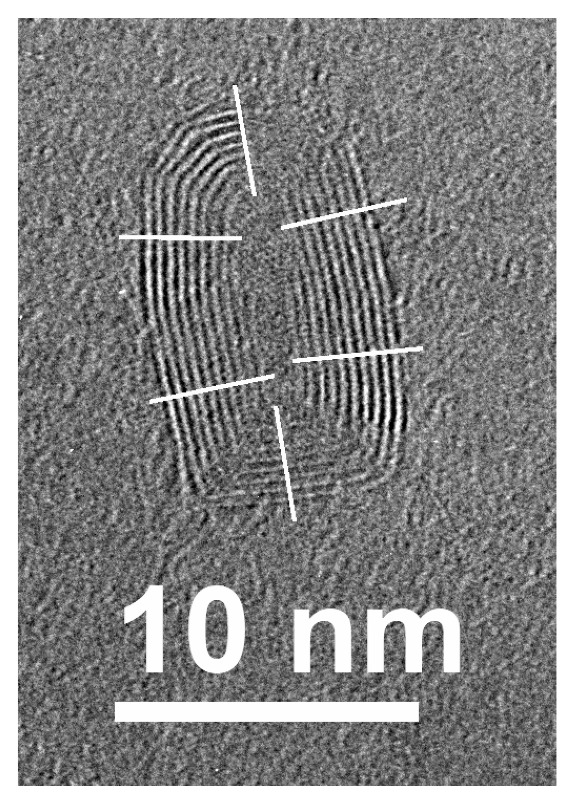
The HRTEM images of a vertical nanotube embedded in the FLG films irradiated with 167 MeV Xe ions at the dose of 5 × 10^11^ ion/cm^2^. The interlayer distances were estimated as 0.34–0.35 nm along white lines on the image and 0.19–0.21 nm for the internal nanocrystal part.

**Figure 7 materials-16-01391-f007:**
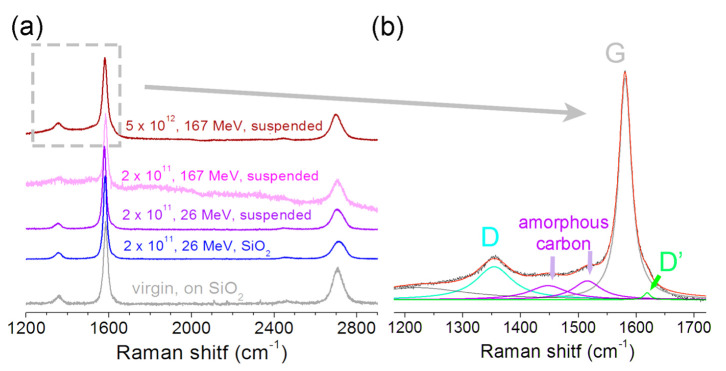
The Raman spectra for pristine and irradiated FLG films transferred before the irradiation onto the SiO_2_/Si substrate or TEM grid, labeled as “on SiO_2_” or “suspended”, respectively. The dose and ion energy are shown as the first and the second parameters in the image, the type of substrate as the third one. (**b**) The spectra deconvolution into components for the CVD FLG film transferred onto the TEM grid and irradiated by 167 MeV Xe ions with dose at 5 × 10^12^ ion/cm^2^. (**a**) A gray dashed square is added to (**a**) to indicate the part shown in (**b**).

**Table 1 materials-16-01391-t001:** The results of Raman spectra fitting for virgin and irradiated CVD FLG films.

Peak Name and Position, cm^−1^	Peaks Area, %		Interpretation
	Virgin	Irradiated ^1^	
G-peak, 1580–1584	87	57	sp^2^, difference between the G-peak positions can be caused by internal mechanical stress in the films.
D-peak, 1354–1355	13	11	sp^3^
1447–1451	-	21	Amorphous carbon modes having both sp^2^ and sp^3^ hybridization, defects and heteroatoms
1515–1518	-	9	
D’-peak, 1618–1619	-	2	A peak is observed for multigraphene near the edges [39] or locally stressed regions

^1^ 167 MeV Xe ions, dose of 5 × 10^12^ ion/cm^2^.

## Data Availability

Data is contained within the article and Appendix A.

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
