# Peer review of "Visualization of Swift Ion Tracks in Suspended Local Diamondized Few-Layer Graphene"

_materials, 2023, doi:10.3390/ma16041391_

Round 1

Reviewer 1 Report

Dear Editor,

The authors have experimentally studied the ion track in suspended FLGs. The manuscript needs to be greatly revised to reach a criterion for receiving an acceptance. So, the following comments should be addressed point-to-point.

1. The novelty of the work is not clear. This issue should be discussed in the introduction section.

2. The excellent features of graphene should be highlighted in the introduction. Some applications of graphene layers should be also given for presenting the role of graphene in different works. As a result, readers could be attracted to the raised applications. The following articles can be assisted to enhance the introduction section so adding those is recommended.

https://doi.org/10.1016/j.optlastec.2022.108680

https://doi.org/10.1016/j.optlastec.2022.108436

https://doi.org/10.3390/cryst9090461

https://doi.org/10.1364/AO.449123

https://doi.org/10.1016/j.diamond.2022.109594

3. What is the used criterion for the amount of dose irradiation? This issue needs to be discussed.

4. Could you explain the amount of surface conductivity for different dose irradiations?

5. The number of samples for measurement should be given in the text.

6. Could you present images for evaluation of the temperature effect in this study?

7. Advantages of this study for use in some works have not been given. Could you explain this issue?

Mohammad Soroosh

Professor of electronics

Shahid Chamran University of Ahvaz

Author Response

  1. The novelty of the work is not clear. This issue should be discussed in the introduction section.

We have added the main novel results (the first visualization of ion tracks in graphene in the form of diamond rings, determination of the main condition for the diamond formation (the absence of a substrate) and estimates of local strains at the track periphery) in the abstract, Introduction and conclusion. Also, please see, the answer on 7 question and the last paragraph of the Discussion.

  1. The excellent features of graphene should be highlighted in the introduction. Some applications of graphene layers should be also given for presenting the role of graphene in different works. As a result, readers could be attracted to the raised applications. The following articles can be assisted to enhance the introduction section so adding those is recommended.

https://doi.org/10.1016/j.optlastec.2022.108680

https://doi.org/10.1016/j.optlastec.2022.108436

https://doi.org/10.3390/cryst9090461

https://doi.org/10.1364/AO.449123

https://doi.org/10.1016/j.diamond.2022.109594

We have added a new paragraph, discussing the application of 2D diamonds and graphene-based composite or heterostructure and including some of these Refs, in the Introduction. Only a part of the above articles was available for reading.

  1. What is the used criterion for the amount of dose irradiation? This issue needs to be discussed.

We have used ion doses lower the typical value for overlapping of the ion tracks usual semiconductor materials. For many materials, it is about ~1012 ion/cm-2. Graphene has a high thermal conductivity, so we reduce the dose used. The highest ion dose used is 5x1011 ion/cm-2, which corresponds to an average distance between ion tracks of 14 nm (dose of 1011 ion/cm-2 corresponds to 32 nm). We have added this explanation at page 3.

  1. Could you explain the amount of surface conductivity for different dose irradiations?

It was not possible to measure conductivity of the suspended graphene, when graphene films are located on Cu grids for HRTEM. When graphene was placed on SiO2/Si substrates we performed a lot of sheet resistance measurements with use of a 4-probe head. The pristine non-irradiated films have a resistance of 1-2 kOhm/sq. After irradiation with Xe in the dose range 1010 – 1011 cm-2, graphene resistivity increases by 10 -15% for the ion energy 167 MeV and by 30-40% for the 26 MeV. More pronounced changes were observed for carrier mobility. For the ion energy of 167 MeV, the electron mobility decreases from ~1200 to ~800  cm2/Vs and for ion energy of 26 MeV to ~10 cm2/Vs. Some data are given in N.A. Nebogatikova, et. al., Nanoscale, 2018, 10(30), 14499.

  1. The number of samples for measurement should be given in the text.

We have used ~4-5 samples for each ion energy and dose. The total quantity of samples was 28 samples.

  1. Could you present images for evaluation of the temperature effect in this study?

Strong local heating of graphene on a SiO2/Si substrate (up to 1500 – 3000oC) takes place in the picosecond time-scale (10-12 – 10-11) [M. Toulemonde, C. Dufour, A. Meftah and E. Paumier, Nucl. Instrum. Methods Phys. Res. Sect. B Beam Interact. Mater. At., 2000, 166–167, 903–912.]. In Ref. [N.A. Nebogatikova, et,  Nanoscale, 2018, 10(30), 14499] the temperature evolution for three-layered graphene is carried out.

Temperature evolution at different radial distances (inset in nanometers) from the ion axis in SiO2. The initial temperature is 300 K, the beam energy 1.3 MeV/amu (for Xe the energy is 170 MeV). M. Toulemonde NIM, 2000

Temperature evolution at different radial distances (inset in nanometers) from the ion axis in SiO2. The initial temperature is 300 K, the beam energy 1.3 MeV/amu (for Xe the energy is 170 MeV). M. Toulemonde NIM, 2000

Temperature and structural changes of three-layered graphene in simulations after the Xe ion irradiation with an energy of 100 MeV.  N.A. Nebogatikova, et al,  Nanoscale, 2018

Temperature and structural changes of three-layered graphene in simulations after the Xe ion irradiation with an energy of 100 MeV.

N.A. Nebogatikova, et,  Nanoscale, 2018

Thus high temperatures (2000-5000 K) are realized for short time, when FLG films are irradiated. The heating time is too short, the term “temperature” is usually used for t > 10-12 s. The temperature range is mentioned now on page 11.

  1. Advantages of this study for use in some works have not been given. Could you explain this issue?

We have obtained structures that cannot be created except by irradiation with high-energy ions (few-layer graphene with nanodiamond rings). The details of the structures of graphene/nanodiamond boundaries and layers properties have not yet been sufficiently understood, but such structures can be considered as a way of bond reconstruction between layers at these interfaces. These carbon structures can be considered as superlattices of joined nanotube and few layer graphene fragments. Their stability and properties are the purpose of study. As it was demonstrated, for example in theoretical studies of Chernozatonskii [C 2021, 7(1), 17; https://doi.org/10.3390/c7010017, Phys. Chem. Chem. Phys., 2016,18, 27432, https://doi.org/10.1039/C6CP05082D] these materials have great perspectives for applications. The electronic band structure of the theoretically predicted nanomesh can be varied from metallic to semiconductor, depending on the interphase structure, size, shape and distance between 2D diamond-like inclusions. Out results are an important step towards creation of these novel superlattice structures. We have added this information to the end of the Discussion.

Reviewer 2 Report

Referee's Comments on:

Visualization of swift ion tracks in suspended local diamondized few-layer graphene

(by Nadezhda A. Nebogatikova et al; submitted to Materials (MDPI) and registered

 as # MATER-23-2140010)

      The manuscript by N.A. Nebogatikova et al reports on the preparation and structural characterisation of a specially modified graphene, which is one of the most popular carbonaceous materials in the science of XXI. century. Already this brief specification of this pivotal topic shows that the respective manuscript can be considered for potential publication in Materials (MDPI).

       Regarding the manuscript as such, it is written well, although some parts have yet some insufficiencies, especially certain commentaries and/or expressions for experiments made (for details, see below). By summing up, this paper can be recommended for publication after some revision, reflecting the following points, mainly the first and last ones:

Addressed Comments:

 1) Abstract; page 1, as such  // Critical comment + Recommendation … I would guess that some readers of this paper will be interested in the purpose and applicability of such specially modified graphene, knowing it already from the abstract that represents the article in all archiving databases (Chemical Abstracts, Web of Science, Scopus, etc.) thus, I would recommend adding a brief information in this sense at the very end of the abstract.

 2) Section 1 (Introduction); page 2, line 52  // Formal note … Better "such a process" or "such a way" than the original incorrect formulation "by such approach".

 3) Section 2; page 2, line 87 // Formal note … As above. What means "sample creation"? Please, find a better expression.

 4) Section 3; page 4, line 140 // Formal note … As in point (2). In context with the text given, better "We identified" or "We revealed" than the originally written "interpreted". This is because an interpretation comes after a finding anything…

 5) Section 3; page 7, lines 216-217 // Critical comment: Question … To my knowledge, diamond is an allotropic form of carbon crystallising in the cubic structure, which gives it it typical hardness. And a hexagonal structuring is being ascribed to the graphite form. Thus, I do not understand why the authors reports here on "hexagonal diamondisation". Please, either explain this discrepancy or correct it (!)

 6) Section 3; page 8, Fig. 6 and its legend (line 237) // Critical comment: Question … The HRTM image shown here is quite impressive. However, what means "vertical nanotube"? What I see on this figure is a "coil of nanotube(s)" or "coiled nanotube(s)". So, the authors should consider renaming their controversial term into that one I propose.

 7) Section 5; pages 11-12, as such // Critical comment … As already mentioned in point (1), this Conclusion misses any general statement on the significance and applicability of so diamondized graphene in scientific practice.

More specifically – where so specially modified graphene may find its use? In material engineering, in electronics, in industrial catalysed productions, in electrochemistry (for manufacturing of new electrodes)? And what are pros and cons of such material(s) in comparison with (i) the bare graphene and (ii) similarly modified graphene already reported in previous papers? All these questions should be answered – best, in a newly added paragraph at the very end of Conclusion. In my opinion, the overall quality of this otherwise interesting and valuable paper would be bettered by this way, approaching it to a wider scientific community.

Author Response

Reviewer 2

  • Abstract; page 1, as such // Critical comment + Recommendation … I would guess that some readers of this paper will be interested in the purpose and applicability of such specially modified graphene, knowing it already from the abstract that represents the article in all archiving databases (Chemical Abstracts, Web of Science, Scopus, etc.) thus, I would recommend adding a brief information in this sense at the very end of the abstract.

OK we have added the aim explonation

  • Section 1 (Introduction); page 2, line 52 // Formal note … Better "such a process" or "such a way" than the original incorrect formulation "by such approach".

Thank you so much for your help

  • Section 2; page 2, line 87 // Formal note … As above. What means "sample creation"? Please, find a better expression.

OK

  • Section 3; page 4, line 140 // Formal note … As in point (2). In context with the text given, better "We identified" or"We revealed" than the originally written "interpreted". This is because an interpretation comes after a finding anything…

Thanks a lot, we have corrected it

  • Section 3; page 7, lines 216-217 // Critical comment: Question … To my knowledge, diamond is an allotropic form of carbon crystallising in the cubic structure, which gives it it typical hardness. And a hexagonal structuring is being ascribed to the graphite form. Thus, I do not understand why the authors reports here on "hexagonal diamondisation". Please, either explain this discrepancy or correct it (!)

Based on the data on interplane spacings in nanocrystals, it was found that part of the nanocrystals has lonsdeilite (or wurtzite) structure. It is also known as the hexagonal diamond (https://www.atomic-scale-physics.de/lattice/struk/hexdia.html), (https://en.wikipedia.org/wiki/Lonsdaleite). Also, some nanoinclusions in the films can be interpreted as an intermediate phase of the transformation from graphene to diamond.

  • Section 3; page 8, Fig. 6 and its legend (line 237) // Critical comment: Question … The HRTM image shown here is quite impressive. However, what means "vertical nanotube"? What I see on this figure is a "coil of nanotube(s)" or "coiled nanotube(s)". So, the authors should consider renaming their controversial term into that one I propose.

The observed interlayer distance of 0.35-0.34 nm and the observed nanoobject size exclude a coiled nanotube version. The minimum internal diameter for SWCNT diameter is about few nm. Therefore, the well-known term "ultrashort nanotubes" can be used. For example, images of ultrashort nanotubes from [Zheyi Chen et al J. AM. CHEM. SOC. 2006, 128, 10568-10571] are shown below. The right image has SWCNTs with diameter of about 1.5-2 nm, and left image demonstrates SWCNTs with diameter > 20 nm.

Images of ultrashort nanotubes from [Zheyi Chen et al J. AM. CHEM. SOC. 2006, 128, 10568-10571]

  • Section 5; pages 11-12, as such // Critical comment … As already mentioned in point (1), this Conclusion misses any general statement on the significance and applicability of so diamondized graphene in scientific practice.

Ultra-short nanotubes, as well as diamond rings, are of interest for applications from the point of view of structuring with the reconstruction of bonds between layers. As it was demonstrated for example in theoretical studies of Chernozatonskii Ref 19,20 [C 2021, 7(1), 17; https://doi.org/10.3390/c7010017, Phys. Chem. Chem. Phys., 2016,18, 27432, https://doi.org/10.1039/C6CP05082D] these materials have high perspectives for applications. One of the structures considered by Chernozatonskii is given in the figure below. The electronic band structure of the theoretically predicted nanomesh can have characteristics ranging from metallic to semiconductor depending on the interphase structure, size, shape and distance between 2D diamond-like inclusions. The film structure is given in Fig. below. Our results is an important step towards the creation of these novel superlattice structures. We have added this information to the Introduction and end of the manuscript.

One of the structures considered by Chernozatonskii

  1. G. Kvashnin, P. Vancsó, L. Y. Antipina, G. I. Márk,L. P. Biró, P. B. Sorokin, and L. A. Chernozatonskii, Nano Res. 8, 1250 (2015).

More specifically – where so specially modified graphene may find its use? In material engineering, in electronics, in industrial catalysed productions, in electrochemistry (for manufacturing of new electrodes)? And what are pros and cons of such material(s) in comparison with (i) the bare graphene and (ii) similarly modified graphene already reported in previous papers? All these questions should be answered – best, in a newly added paragraph at the very end of Conclusion. In my opinion, the overall quality of this otherwise interesting and valuable paper would be bettered by this way, approaching it to a wider scientific community.

From our previous paper [N.A. Nebogatikova, et,  Nanoscale, 2018, 10(30), 14499]:

“The morphology and electronic properties of single and few-layer graphene films nanostructured by the impact of heavy high-energy ions leads to the formation of nano-sized pores, or antidots, with sizes ranging from 20 to 60 nm, in the upper one or two layers. With increasing ion energy (>70 MeV), a profound reduction in the concentration of structural defects (by a factor of 2–5), relatively high mobility values of charge carriers (700–1200 cm2/Vs) and a transport band gap of about 50 meV were observed in the nanostructured films.” We also found that the formation of “welded” sheets due to interlayer covalent bonds at the edges and, hence, defect-free antidot arrays is likely at high ion energies (above 70 MeV). The results are connected with the partial interlayer bonds recovery. Formation of the diamond rings can lead to interlayer reconstruction. We plan to create the suspended FLG films, to irradiate them and different regimes and to study electrical properties. But now it is only in progress.”

Round 2

Reviewer 1 Report

Dear Editor,

I studied the manuscript again and the response file. The manuscript has been enhanced in its present form and it can be recommended for publication.

Kind regards,

Mohammad Soroosh

Professor of Electronics

Shahid Chamran University of Ahvaz, Iran